# Provably Robust Federated Reinforcement Learning

## Abstract

Federated reinforcement learning (FRL) allows agents to jointly learn a global decision-making policy under the guidance of a central server. While FRL has advantages, its decentralized design makes it prone to poisoning attacks. To mitigate this, Byzantine-robust aggregation techniques tailored for FRL have been introduced. Yet, in our work, we reveal that these current Byzantine-robust techniques are not immune to our newly introduced Normalized attack. Distinct from previous attacks that targeted enlarging the distance of policy updates before and after an attack, our Normalized attack emphasizes on maximizing the angle of deviation between these updates. To counter these threats, we develop an ensemble FRL approach that is provably secure against both known and our newly proposed attacks. Our ensemble method involves training multiple global policies, where each is learnt by a group of agents using any foundational aggregation rule. These well-trained global policies then individually predict the action for a specific test state. The ultimate action is chosen based on a majority vote for discrete action systems or the geometric median for continuous ones. Our experimental results across different settings show that the Normalized attack can greatly disrupt non-ensemble Byzantine-robust methods, and our ensemble approach offers substantial resistance against poisoning attacks.

## 1 Introduction

**Background and Motivation:** Reinforcement learning (RL) is a sequential decision-making procedure and can be modeled as a Markov decision process (MDP) [35]. Specifically, an agent in RL operates by taking actions according to a certain policy within a stochastic environment. The agent earns rewards for its actions and uses these rewards to enhance its policy. The ultimate goal of the agent is to learn the best possible policy by consistently engaging with the environment, aiming to maximize its cumulative rewards over the long term. Despite the advancements in current RL models, they are often data-intensive and face issues due to their limited sample efficiency [12, 13]. To tackle this challenge, a straightforward solution might be parallel RL [27, 29]. In parallel RL, agents share their trajectories with a central server to train a policy. However, this is often impractical due to high communication costs, especially for IoT devices [40], and prohibited in applications like medical records [23] due to data sensitivity.

Toward this end, the federated reinforcement learning (FRL) [13, 15–17, 22, 45] paradigm has been introduced as a solution to the problems faced by traditional parallel RL methods. In FRL, various agents work together to train a global policy under the guidance of a central server, all while keeping their raw trajectories private. Specifically, during each global training round, the central server shares the current global policy with all agents or a selected group. Agents then refine their local policies using the shared global policy and through interactions with their environment. Subsequently, agents send their local policy updates back to the server. Once the server receives these policy updates from the agents, it employs aggregation rule, like FedAvg [25], to merge these received policy updates and refine the global policy. Owing to its willingness to respect agents' privacy, FRL has been widely deployed in real-world systems, such as robotics [18], autonomous driving [20], and IoT network [40].

While FRL has its merits, it is susceptible to poisoning attacks owing to its decentralized nature [13]. Such an attack might involve controlling malicious agents, who may either corrupt their local training trajectories (known as *data poisoning attacks* [13]), or intentionally send carefully crafted policy updates to the server (known as *model poisoning attacks* [3, 14, 34]), with an aim to manipulate the ultimately learnt global policy. A seemingly direct defense against these poisoning attacks would be to implement existing federated learning (FL) based aggregation rules, such as Trimmed-mean [44] and Median [44], within the FRL context. Nevertheless, as subsequent experimental results will demonstrate, merely extending existing FL-based aggregation rules does not provide a satisfactory defense performance. This is because these rules, originally designed for FL, remain vulnerable to poisoning attacks [14, 34]. Within the domain of FRL, a recently introduced Byzantine-robust aggregation rule, FedPG-BR [13], has demonstrated exceptional robustness against existing advanced poisoning attacks [14, 34].

**Our work:** In this paper, we propose the first model poisoning attacks to Byzantine-robust FRL. In the attack we propose, the attacker deliberately crafts the policy updates on malicious agents so as to maximize the discrepancy between the aggregated policy updates before and after the attack. While a direct strategy might be to maximize the distance between the aforementioned policy updates [34], this method only accounts for the magnitude of the aggregated policy update, neglecting its directionality. To address this challenge, we propose the *Normalized attack*, wherein the attacker strives to maximize the angular deviation between the aggregated policy updates pre and post-attack. Nevertheless, solving the reformulated optimization problem remains challenging, especially given that the existing robust aggregation rules like FedPG-BR [13] are not differentiable. To tackle this issue, we introduce a two-stage approach to approximate the solution to the optimization problem. Specifically, in the first stage, we determine the optimal direction for the malicious policy updates, and in the second phase, we calculate the optimal magnitude for these malicious policy updates.

We subsequently propose an innovative *ensemble FRL* method that is provably secure against both existing attacks and our newly proposed Normalized attack. Within our proposed ensemble framework, we first leverage a deterministic method to divide agents into multiple non-overlapping groups by using the hash values of the agents' IDs. Each group then trains a global policy, employing a *foundational aggregation rule* such as Median [44] and FedPG-BR [13], using the agents within its respective group. During the testing phase, given a test state $s$, we deploy the well-trained multiple global policies to predict the action for state $s$. Considering that the action space in FRL may be either discrete or continuous, we apply varying strategies to aggregate these predicted actions

accordingly. Specifically, in a discrete action space, we select the action with the highest frequency as the final action. Conversely, if the action space is continuous, the final action is determined by calculating the geometric median [9] of the predicted actions. We theoretically prove that our proposed ensemble method will consistently predict the same action for the test state $s$ before and after attacks, provided that the number of malicious agents is below a certain threshold when the action space is discrete. In the context of a continuous space FRL system, we demonstrate that the distance between actions predicted by our ensemble approach, before and after the attack, is bounded, as long as the number of malicious agents is less than half of the total number of groups.

Our proposed Normalized attack and the proposed ensemble method have been thoroughly evaluated on three RL benchmark datasets. These include two discrete datasets, namely Cart Pole [2] and Lunar Lander [11], and one continuous dataset, Inverted Pendulum [2]. We benchmarked against four existing poisoning attacks including Random action attack [13], Random noise attack [13], Trim attack [14], and Shejwalkar attack [34]. Furthermore, we employed six foundational aggregation rules for evaluation including FedAvg [25], Trimmed mean [44], Median [44], geometric median [9], FLAME [30], and FedPG-BR [13]. Experimental findings illustrate that our proposed Normalized attack can remarkably manipulate non-ensemble-based methods (where a single global policy is learnt using all agents along with a particular foundational aggregation rule). Distinctively, within a non-ensemble context, our Normalized attack stands out as the exclusive poisoning attack that can target the FRL-specific aggregation rule. We further demonstrate that our proposed ensemble method can effectively defend against all considered poisoning attacks, including our Normalized attack. Notably, for all robust foundational aggregation rules, the test reward of our proposed ensemble method, even when under attack, closely mirrors that of the FedAvg in a non-attack scenario. Our main contributions can be summarized as follows:

- We propose the Normalized attack, the first model poisoning attacks tailored to Byzantine-robust FRL.
- We propose an efficient ensemble FRL method that is provably secure against poisoning attacks.
- Comprehensive experiments highlight that our proposed Normalized attack can notably compromise non-ensemble-based robust foundational aggregation rules. Additionally, our proposed ensemble method shows significant capability in defending against both existing and our newly introduced poisoning attacks.

## 2 Preliminaries and Related Work

### 2.1 Federated Reinforcement Learning

A federated reinforcement learning (FRL) system [13, 16, 17] consists of $n$ agents and a central server collaborating to train a global policy. Each agent $i \in [n]$ solves a local Markov decision process (MDP) [35], defined as $\mathcal{M}_i = \{\mathcal{S}, \mathcal{A}, \mathcal{P}_i, \mathcal{R}_i, \gamma_i, \rho_i\}$, with $\mathcal{S}$ as the state space, $\mathcal{A}$ the action space, $\mathcal{P}_i$ the transition probability, $\mathcal{R}_i$ the reward function, $\gamma_i$ the discount factor, and $\rho_i$ the initial state distribution. In FRL, agent $i$ follows a policy $\pi$ that gives the probability of taking action $a$ in state $s$. Through interactions with its environment, the agent generates a trajectory

$\tau_i = \{s_{i,1}, a_{i,1}, ..., s_{i,H}, a_{i,H}\}$, starting from an initial state $s_{i,1}$ drawn from $\rho_i$, with $H$ as the trajectory length. The cumulative reward is calculated as $\mathcal{R}(\tau_i) = \sum_{h \in [H]} \gamma_i^h \mathcal{R}_i(s_{i,h}, a_{i,h})$. Let $\pi_\theta$ denote a policy parameterized by $\theta \in \mathbb{R}^d$, where $d$ is the dimension of $\theta$. The distribution of agent $i$'s trajectories under $\pi_\theta$ is $p_i(\tau_i|\pi_\theta)$. For simplicity, we will refer to $\theta$ as $\pi_\theta$. Agent $i$ evaluates the effectiveness of a policy $\pi$ by solving the following optimization problem:

$$J_i(\theta) = \mathbb{E}_{\tau_i \sim p_i(\cdot|\theta)}[\mathcal{R}(\tau_i)|\mathcal{M}_i]. \tag{1}$$

In FRL, the $n$ agents collaborate to train a global policy aimed at maximizing the total cumulative discounted reward. Thus, the optimization problem in FRL becomes $\max_{\theta \in \mathbb{R}^d} \sum_{i \in [n]} J_i(\theta)$. FRL solves this problem in an iterative manner. Specifically, in each global training round $t$, FRL performs the following three steps:

- **Step I: Global policy synchronization.** The server distributes the current global policy $\theta$ to all agents or a selection of them.
- **Step II: Local policy updating.** Each agent $i \in [n]$ uses the current policy $\theta$ to sample a batch of trajectories $\{\tau_i^k\}_{k=1}^B$, where $\tau_i^k = \{s_{i,1}^k, a_{i,1}^k, s_{i,2}^k, a_{i,2}^k, \ldots, s_{i,H}^k, a_{i,H}^k\}$, $B$ is the batch size. Subsequently, agent $i$ calculates a local policy update $g_i$. For example, using the REINFORCE algorithm [41], $g_i$ is calculated as:

$$g_i = \frac{1}{B} \sum_{k \in [B]} \left[ \sum_{h \in [H]} \nabla_\theta \log \pi_\theta(a_{i,h}^k|s_{i,h}^k) \right] \times$$
$$\left[ \sum_{h \in [H]} \gamma_i^h \mathcal{R}_i(s_{i,h}^k, a_{i,h}^k) - \mathfrak{I} \right], \tag{2}$$

where $\mathfrak{I}$ is a constant. Then agent $i$ sends $g_i$ to the server.

- **Step III: Global policy updating.** The server updates the global policy by aggregating local updates using AR$\{\cdot\}$:

$$\theta = \theta + \eta \cdot \text{AR}\{g_i : i \in [n]\}, \tag{3}$$

where $\eta$ is the learning rate.

FRL methods vary in their aggregation rules. For example, using FedAvg [25], the global policy is updated as: $\theta = \theta + \frac{\eta}{n} \sum_{i \in [n]} g_i$.

### 2.2 Poisoning Attacks to FRL

The distributed nature of FRL makes it susceptible to poisoning attacks [13, 14, 34], where malicious agents manipulate local training data (data poisoning) or policy updates (model poisoning) to compromise the global policy. For example, in Random action attack [13], agents act randomly without following a pattern. Model poisoning attacks include Random noise attack [13], where agents send Gaussian noise as policy updates, Trim attack [14], which maximizes deviation in policy updates, and Shejwalkar attack [34], which increases the distance between pre- and post-attack updates. While some studies [24, 47] assume agents can manipulate environments or rewards, such scenarios are often impractical and are not considered in our paper.

### 2.3 Byzantine-robust Aggregation Rules

*2.3.1 FL-based Aggregation Rules.* In typical federated learning (FL), the server uses FedAvg [25] to aggregate local model updates[1],

---

[1]Note that in FL, we commonly refer to a "model update" rather than a "policy update".

but this method is vulnerable to poisoning attacks since even one malicious agent can skew the results. To counter such attacks, several Byzantine-resilient aggregation rules have been proposed [4, 7–9, 28, 30–33, 43, 44, 48]. Examples include Median [44], which computes the median for each dimension, and Trimmed-mean [44], which removes extreme values before averaging. FLAME [30] clusters agents based on cosine similarity, discarding suspicious updates and adding adaptive noise to the rest. Our proposed ensemble method differs from [8] by addressing continuous action spaces, while their approach only supports categorical labels. We also provide theoretical evidence that an attacked agent behaves similarly to pre-attack conditions as long as malicious agents are fewer than half of the total groups.

*2.3.2 FRL-based Aggregation Rules.* The authors in [13] proposed FedPG-BR to defend against poisoning attacks in FRL. Each training round, the server computes the vector median of local policy updates and marks an update as benign if it aligns in direction and magnitude with the median. It then averages these benign updates to form a policy update estimator. Additionally, the server samples trajectories to compute its own policy update. The final global policy update is obtained by combining the estimator and the server's update using the stochastically controlled stochastic gradient (SCSG) [19] to reduce variance.

**Table 1: Comparison among the Trim attack, Shejwalkar attack, and our proposed Normalized attack.**

|  | Direction | Magnitude |
|---|:---:|:---:|
| Trim attack [14] | ✗ | ✗ |
| Shejwalkar attack [34] | ✗ | ✓ |
| Normalized attack | ✓ | ✓ |

*2.3.3 Limitations of Existing Attacks and Defenses.* Current poisoning attacks and defense strategies have limitations. The Trim attack [14] targets individual dimensions in linear aggregation rules like Trimmed-mean and Median [44], ignoring the update's overall direction. In contrast, the Shejwalkar attack [34] considers the entire update but overlooks its direction. Table 1 compares these with our proposed Normalized attack. Additionally, applying FL-based aggregation rules in FRL leads to poor performance, as they remain vulnerable to known attacks [14, 34]. Although FedPG-BR [13] counters Trim and Shejwalkar attacks, our experiments show it is still vulnerable to our Normalized attack.

## 3 Problem Setting

**Threat model:** We adopt the threat model from [13], where an attacker controls some malicious agents. These agents may poison their training trajectories or send random policy updates to the server. The attacker's goal is to disrupt the global policy's convergence or push it toward a bad optimum. In a full knowledge attack, the attacker knows all agents' policy updates and the server's aggregation rule. In a partial knowledge attack, the attacker only knows the malicious agents' updates and the aggregation rule.

**Defense objectives:** We aim to propose a method that achieves the following two goals. I) Competitive learning performance: In non-adversarial settings, the method should perform as well as FedAvg,

achieving comparable test rewards when all agents are benign. II) Resilience: It should defend against both data and model poisoning attacks. Even under attacks, the final global policy should maintain test rewards similar to those of FedAvg in attack-free scenarios.

## 4 Our Attack

### 4.1 Attack as an Optimization Problem

In our proposed Normalized attack, the attacker crafts malicious policy updates to maximize the deviation between the aggregated updates before and after the attack. A simple way to achieve this is by maximizing the distance between the two updates, resulting in an optimization problem for each global training round:

$$\max \left\| \mathrm{AR}\{g_i : i \in [n]\} - \widehat{\mathrm{AR}}\{g_i : i \in [n]\} \right\|, \quad (4)$$

where $\|\cdot\|$ denotes the $\ell_2$-norm, and $\mathrm{AR}\{g_i : i \in [n]\}$ and $\widehat{\mathrm{AR}}\{g_i : i \in [n]\}$ represent the aggregated policy updates before and after the attack, respectively. However, Eq. (4) focuses only on the magnitude of the post-attack aggregated update, ignoring its direction. As a result, the original and attacked updates could align in the same direction. Since FedPG-BR [13] evaluates both the direction and magnitude of policy updates, attackers must carefully craft updates to bypass this defense. To address this, we propose the *Normalized attack*, which maximizes the angular deviation between the original and attacked aggregated updates, rather than just their magnitude. The formulation of our Normalized attack is as follows:

$$\max \left\| \frac{\mathrm{AR}\{g_i : i \in [n]\}}{\|\mathrm{AR}\{g_i : i \in [n]\}\|} - \frac{\widehat{\mathrm{AR}}\{g_i : i \in [n]\}}{\left\|\widehat{\mathrm{AR}}\{g_i : i \in [n]\}\right\|} \right\|. \quad (5)$$

### 4.2 Solving the Optimization Problem

Solving Problem (5) is challenging because many aggregation rules, like FedPG-BR [13], are non-differentiable. To overcome this, we use practical techniques to approximate the solution by determining the direction of malicious updates in Stage I, followed by calculating their magnitude in Stage II.

**Stage I (Optimize the direction):** We let $\mathcal{B}$ be the set of malicious agents. Assume that the malicious policy update $g_j$, $j \in \mathcal{B}$, is the perturbed version of normalized benign policy update:

$$g_j = \frac{\mathrm{AR}\{g_i : i \in [n]\}}{\|\mathrm{AR}\{g_i : i \in [n]\}\|} + \lambda\Delta, \quad j \in \mathcal{B}, \quad (6)$$

where $\lambda$ is an adjustment parameter and $\Delta$ is a perturbation vector. Then we can reformulate Problem (5) as follows:

$$\underset{\lambda,\Delta}{\mathrm{argmax}} \left\| \frac{\mathrm{AR}\{g_i : i \in [n]\}}{\|\mathrm{AR}\{g_i : i \in [n]\}\|} - \frac{\widehat{\mathrm{AR}}\{g_i : i \in [n]\}}{\left\|\widehat{\mathrm{AR}}\{g_i : i \in [n]\}\right\|} \right\|$$
$$\text{s.t.} \quad g_j = \frac{\mathrm{AR}\{g_i : i \in [n]\}}{\|\mathrm{AR}\{g_i : i \in [n]\}\|} + \lambda\Delta, \quad j \in \mathcal{B}. \quad (7)$$

Finding the optimal $\lambda$ and $\Delta$ simultaneously is also not trivial. In this paper, we fix the $\Delta$ and turn to finding the optimal $\lambda$. For example, we can let $\Delta = -\mathrm{sign}(\mathrm{Avg}\{g_i : i \in [n]\})$, where $\mathrm{Avg}\{g_i : i \in [n]\}$ means the average of $n$ local policy updates. After we fix

$\Delta$, the optimization problem of Eq. (7) becomes the following:

$$\underset{\lambda}{\text{argmax}} \left\| \frac{\text{AR}\{g_i : i \in [n]\}}{\|\text{AR}\{g_i : i \in [n]\}\|} - \frac{\widehat{\text{AR}}\{g_i : i \in [n]\}}{\left\|\widehat{\text{AR}}\{g_i : i \in [n]\}\right\|} \right\|$$

$$\text{s.t.} \quad g_j = \frac{\text{AR}\{g_i : i \in [n]\}}{\|\text{AR}\{g_i : i \in [n]\}\|} + \lambda\Delta, \quad j \in \mathcal{B}. \tag{8}$$

There exist multiple methods to determine $\lambda$. In this study, we adopt the subsequent way to compute $\lambda$. Specifically, in each global training round, if the value of $\left\| \frac{\text{AR}\{g_i:i\in[n]\}}{\|\text{AR}\{g_i:i\in[n]\}\|} - \frac{\widehat{\text{AR}}\{g_i:i\in[n]\}}{\left\|\widehat{\text{AR}}\{g_i:i\in[n]\}\right\|} \right\|$ increases, then we update $\lambda$ as $\lambda = \lambda + \hat{\lambda}$, otherwise we let $\lambda = \lambda - \hat{\lambda}$. We repeat this process until the convergence condition satisfies, e.g., the difference of $\lambda$ between two consecutive iterations is smaller than a given threshold.

**Stage II (Optimize the magnitude):** After obtaining the direction of malicious policy update $g_j$, we proceed to demonstrate how to determine the magnitude of $g_j$ for $j \in \mathcal{B}$. In particular, let $\tilde{g}_j$ represent the scaled policy update for malicious agent $j$, where $\tilde{g}_j = \frac{g_j}{\|g_j\|} \times \zeta$, and $\zeta$ is the scaling factor. We then formulate the following optimization problem to determine the scaling factor $\zeta$:

$$\underset{\zeta}{\text{argmax}} \left\| \frac{\text{AR}\{g_i : i \in [n]\}}{\|\text{AR}\{g_i : i \in [n]\}\|} - \frac{\widehat{\text{AR}}\{g_i : i \in [n]\}}{\left\|\widehat{\text{AR}}\{g_i : i \in [n]\}\right\|} \right\|$$

$$\text{s.t.} \quad \tilde{g}_j = \frac{g_j}{\|g_j\|} \times \zeta, \quad j \in \mathcal{B}. \tag{9}$$

The way to compute $\zeta$ is similar to that of $\lambda$. Specifically, if $\left\| \frac{\text{AR}\{g_i:i\in[n]\}}{\|\text{AR}\{g_i:i\in[n]\}\|} - \frac{\widehat{\text{AR}}\{g_i:i\in[n]\}}{\left\|\widehat{\text{AR}}\{g_i:i\in[n]\}\right\|} \right\|$ increases, we update $\zeta$ as $\zeta = \zeta + \hat{\zeta}$, otherwise $\zeta = \zeta - \hat{\zeta}$. We repeat this process until the convergence condition is met, then malicious agent $j$ sends $\tilde{g}_j$ to the server.

Fig. 1 shows the impact of our Normalized attack. In each global round, the attacker maximizes the deviation between pre- and post-attack aggregated updates, causing the global policy $\theta$ to drift. Over multiple rounds, this drift leads the FRL system to converge to a suboptimal solution. Since RL loss functions are highly non-convex, with many local optima, the attack's impact can be significant.

Note that we do not provide a theoretical analysis of our attack for the following reasons: In our Normalized attack, the attacker carefully crafts malicious updates to induce subtle deviations in the aggregated policy each round. These deviations are hard to detect but still degrade the model's performance. Modeling them theoretically is challenging. As shown in prior works [14, 34], the true goal of an attack is its real-world impact, such as causing incorrect predictions or compromising security. While theory offers insights, practical performance better reflects real-world outcomes.

## 5 Our Defense

### 5.1 Overview

In our approach, we train multiple global policies instead of a single one, each using a foundational aggregation rule like Trimmed-mean or Median [44] with different subsets of agents. During testing, the agent predicts an action using all trained policies. For discrete

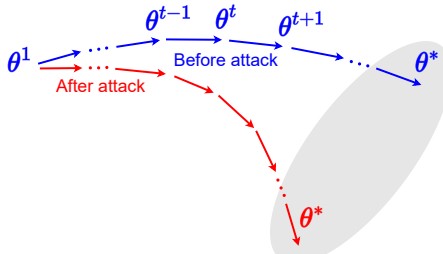

**Figure 1: Illustration of the effects of our Normalized attack. $\theta^1$ is the initial global policy, $\theta^*$ is a local optimum.**

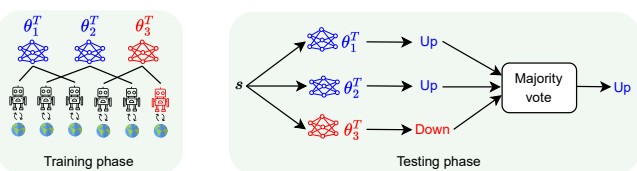

**Figure 2: Illustration of our ensemble framework with discrete action space.**

action spaces, the final action is chosen by majority vote, while for continuous spaces, it is determined by the geometric median [9]. Fig. 2 illustrates the process for a discrete action space with six agents split into three groups, each training a global policy over $T$ rounds, resulting in policies $\theta_1^T$, $\theta_2^T$, and $\theta_3^T$. Since the third group contains a malicious agent, $\theta_3^T$ is poisoned. During evaluation, given a test state $s$, the three policies predict "UP", "UP", and "Down". With a majority vote, the final action selected is "UP".

### 5.2 Our Ensemble Method

In an FRL system with $n$ agents, our method divides them into $K$ non-overlapping groups deterministically, such as by hashing their IDs. Each group trains a global policy using its agents with an aggregation rule like Trimmed-mean, Median [44], or FedPG-BR [13]. Let $\theta_k^T$ represent the policy learned by group $k$ after $T$ global rounds. At the end of training, we obtain $K$ global policies: $\theta_1^T, \theta_2^T, ..., \theta_K^T$. During testing, the agent independently executes the $K$ trained policies. At a test state $s$, let $F(s, \theta_k^T)$ denote the action taken by the agent using policy $\theta_k^T$. With $K$ policies, the agent generates $K$ actions: $F(s, \theta_1^T), F(s, \theta_2^T), \ldots, F(s, \theta_K^T)$. The final action at state $s$ is determined using an ensemble method, which varies based on whether the action space $\mathcal{A}$ is discrete or continuous.

**Discrete action space:** If the action space $\mathcal{A}$ is discrete, the agent's action is determined by majority vote among the $K$ actions. Let $v(s, a)$ represent the frequency of action $a$ at state $s$, calculated as:

$$v(s, a) = \sum_{k \in [K]} \mathbb{1}_{\{F(s, \theta_k^T) = a\}}, \tag{10}$$

where $\mathbb{1}$ is the indicator function, which returns 1 if $F(s, \theta_k^T) = a$, and 0 otherwise. The final action $\Phi(s)$ at test state $s$ is the one with the highest frequency, calculated as:

$$\Phi(s) = \underset{a \in \mathcal{A}}{\text{argmax}} \, v(s, a). \tag{11}$$

**Continuous action space:** For a continuous action space $\mathcal{A}$, we use the Byzantine-robust geometric median [9] to aggregate the $K$

actions. The final action at state $s$ is computed as:

$$\Phi(s) = \arg\min_{a \in \mathcal{A}} \sum_{k \in [K]} \left\| F(s, \theta_k^T) - a \right\|. \tag{12}$$

We use the geometric median [9] to aggregate the $K$ continuous actions instead of FedAvg or Trimmed-mean, as our experiments show that these methods are vulnerable to poisoning attacks.

**Complete algorithm:** Algorithm 1 in Appendix outlines the ensemble method during training. In Lines 4-10, each group trains its global policy in round $t$. In Line 5, the server for group $k$ shares the current global policy with its agents, who refine their local policies and send updates back (Lines 6-8). Here, $n_k$ represents the agents in group $k$. Finally, the server aggregates these updates to revise the global policy (Line 9). Algorithm 2 in Appendix summarizes the testing phase, where the final action is selected by majority vote for discrete actions (Line 3) or by geometric median for continuous actions (Line 5).

**Complexity analysis:** In our ensemble FRL approach, each agent participates in only one global training round over $T$ rounds. Thus, the computational cost per agent is $O(T)$.

## 5.3 Formal Security Analysis

In this section, we present the security analysis of our ensemble method. For discrete action spaces, we show that the predicted action at a test state $s$ remains unchanged despite poisoning attacks, as long as the number of malicious agents stays below a certain threshold. For continuous action spaces, we prove that the difference between actions predicted before and after an attack is bounded if malicious agents make up less than half of the groups.

THEOREM 1 (DISCRETE ACTION SPACE). *Consider an FRL system with $n$ agents and a test state $s$, where the action space $\mathcal{A}$ is discrete. The agents are divided into $K$ non-overlapping groups based on the hash values of their IDs, and each group trains its global policy using an aggregation rule AR. Define actions $x$ and $y$ as those with the highest and second-highest frequencies for state $s$, with ties resolved by selecting the action with the smaller index. Our ensemble method aggregates the $K$ actions using Eq. (11). Let $\Phi(s)$ and $\Phi'(s)$ represent the actions predicted when all agents are benign and when up to $n'$ agents are malicious, respectively. The condition for $n'$ is:*

$$n' = \left\lfloor \frac{v(s, x) - v(s, y) - \mathbb{1}_{\{y < x\}}}{2} \right\rfloor, \tag{13}$$

*where $v(s, x)$ and $v(s, y)$ represent the pre-attack frequencies of actions $x$ and $y$ for state $s$, respectively. The notation $y < x$ indicates that action $y$ has a smaller index than action $x$. Then we have that:*

$$\Phi(s) = \Phi'(s) = x. \tag{14}$$

PROOF. The proof is in Appendix A.1. □

THEOREM 2 (CONTINUOUS ACTION SPACE). *In a continuous action space FRL system with $n$ agents and a test state $s$, the agents are divided into $K$ non-overlapping groups. If $n'$ agents are malicious and $n' < K/2$, each group trains a global policy using an aggregation rule AR. Our ensemble method aggregates the $K$ continuous actions using*

Eq. (12). *Let $\Phi(s)$ and $\Phi'(s)$ be the actions predicted before and after the attack, respectively. The following holds:*

$$\left\| \Phi(s) - \Phi'(s) \right\| \leq \frac{2w(K - n')}{K - 2n'}, \tag{15}$$

*where $w = \max\left\{ \|F(s, \theta_k^T) - \Phi(s)\| : k \in [K] \right\}$, $\{F(s, \theta_k^T) : k \in [K]\}$ is the set of $K$ continuous actions before attack.*

PROOF. The proof is in Appendix A.2. □

REMARK. *Our framework accounts for cases where honest agents may act similarly to malicious ones. Theorems 1 and 2 hold as long as the total number of adversarial agents—both malicious and unintentionally adversarial—stays within a certain limit.*

## 6 Evaluation

### 6.1 Experimental Setup

*6.1.1 Datasets.* We use the following three datasets from different domains, including two discrete datasets (Cart Pole [2], Lunar Lander [11]), and one continuous dataset (Inverted Pendulum [2]). Details of these datasets are provided in Appendix A.4.

*6.1.2 Compared Poisoning Attacks.* We compare our Normalized attack with one data poisoning attack (Random action attack [13]) and three model poisoning attacks (Random noise [13], Trim [14], and Shejwalkar [34]). Details are in Appendix A.5.

*6.1.3 Foundational Aggregation Rules.* We consider the following state-of-the-art foundational aggregation rules, including five aggregation rules designed for FL (FedAvg [25], coordinate-wise trimmed mean (Trimmed-mean) [44], coordinate-wise median (Median) [44], geometric median [9] and FLAME [30]) and one aggregation rule designed for FRL (FedPG-BR [13]). Details are in Appendix A.6.

*6.1.4 Evaluation Metric.* We evaluate an FRL method's robustness using *test reward*. For the non-ensemble approach, the test reward is the average reward from 10 sampled trajectories using the trained global policy. In our ensemble method, we also average rewards from 10 trajectories, but actions are predicted using the ensemble framework. A lower test reward indicates a more effective attack and weaker defense.

*6.1.5 Parameter Settings.* By default, we assume that there are 30 agents in total. Following [13], we assume that 30% of agents are malicious. For our proposed ensemble method, we partition the agents into $K = 5$ disjoint groups. The batch sizes $B$ for Cart Pole, Lunar Lander, and Inverted Pendulum datasets are set to 16, 64, and 32, respectively. The learning rates for these three datasets are individually set to $1 \times 10^{-3}$, $3 \times 10^{-3}$, and $1 \times 10^{-3}$. Furthermore, for each of the three datasets, every agent samples a total of 5,000, 10,000, and 5,000 trajectories during the training phase, respectively. Regarding the policy architectures, we train a Categorical MLP for the Cart Pole and Lunar Lander datasets and a Gaussian MLP for the Inverted Pendulum dataset. The policy architectures are shown in Table 3 in Appendix. We assume all agents use the same discount factor $\gamma$, trajectory horizon $H$, and $\mathfrak{I}$. Our Normalized attack has parameters $\Delta$, $\hat{\lambda}$ and $\hat{\zeta}$. The default value of these six parameters are shown in Table 4 in Appendix. FedPG-BR uses unique parameters, including mini-batch size $b$, global sampling

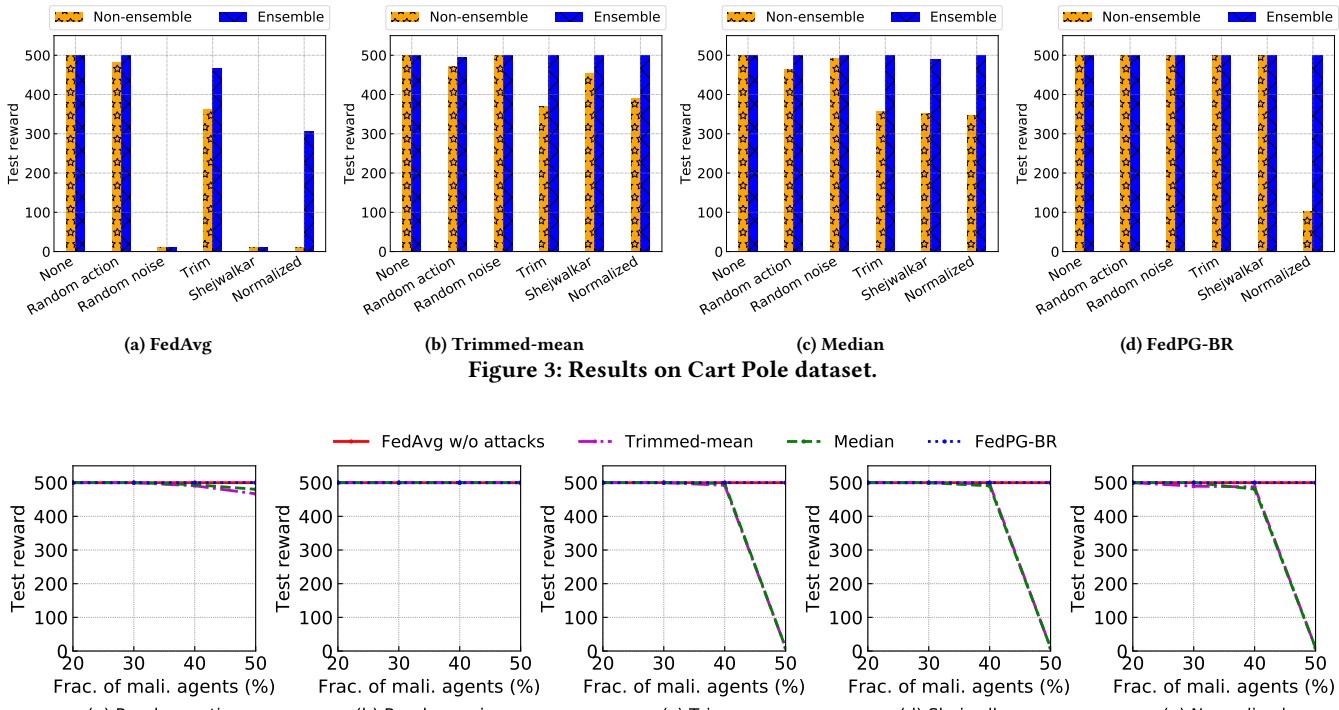

Figure 3: Results on Cart Pole dataset.

Figure 4: Impact of the fraction of malicious agents on our ensemble method, where the Cart Pole dataset is considered.

steps $N$, variance bound $\sigma$, and confidence parameter $\delta$. Detailed settings are in Table 5 in Appendix. We assume the attacker has full knowledge of the FRL system unless stated otherwise. Results are presented on the Cart Pole dataset by default. We compare our ensemble method with the non-ensemble approach. In the non-ensemble method, the server trains a single global model with all agents using a foundational aggregation rule from Section 6.1.3. In our ensemble method, agents are divided into $K$ groups, each training a global policy with the same aggregation rule by default.

## 6.2 Experimental Results

**Normalized attack is effective against non-ensemble methods:** Fig. 3 shows the results of different defenses under various attacks on Cart Pole dataset. The results on Lunar Lander and Inverted Pendulum datasets are shown in Fig. 9 and Fig. 10 in Appendix, respectively. "None" means all agents are benign. Based on Fig. 3 and Figs. 9-10, it is evident that our proposed Normalized attack successfully targets the non-ensemble methods. For instance, in the Lunar Lander dataset, our Normalized attack reduces the test reward of the Median to -33.3 in the non-ensemble setting, compared to a reward of 219.3 when all agents are benign. Notably, our Normalized attack stands out as the sole method capable of substantially manipulating the non-ensemble FedPG-BR aggregation rule across all three datasets. For example, in the Cart Pole dataset, the test reward of non-ensemble-based FedPG-BR drops from 500 in the absence of an attack to 101.4 under our Normalized attack. However, existing attacks such as Trim attack and Shejwalkar attack achieve unsatisfactory attack performance. The reason is that the

Trim attack solely takes into account each dimension of the policy update, neglecting the entirety of the update itself. Furthermore, the Shejwalkar attack ignores the direction of the policy update.

**Our ensemble method is effective:** From Fig. 3 and Figs. 9-10 (in Appendix), we observe that when all agents are benign, our ensemble framework achieves test rewards comparable to FedAvg without attacks across all datasets and Byzantine-robust aggregation rules, fulfilling the goal of "competitive learning performance." For example, in the Inverted Pendulum dataset, the Trimmed-mean rule within the ensemble framework achieves a test reward of 1000, matching FedAvg's performance without attacks. However, non-ensemble Byzantine-robust aggregation rules remain vulnerable to poisoning attacks, including our Normalized attack. As shown in the figures, embedding these robust rules within our ensemble framework ensures defense against all considered attacks, achieving the "resilience" goal. For instance, in the Cart Pole dataset under the Normalized attack, FedPG-BR achieves a test reward of 101.4 in the non-ensemble setting but 500 within our ensemble framework. Similarly, for the Inverted Pendulum dataset, Trimmed-mean yields test rewards of 152.7 without ensemble but 1000 with it under the Random action attack. However, FedAvg, even within our ensemble framework, remains vulnerable to attacks due to its inherent lack of robustness.

**Impact of the fraction of malicious agents:** Fig. 4 shows the impact of the fraction of malicious agents on the robustness of our ensemble method on Cart Pole dataset. From Fig. 4, we observe that when a large fraction of agents are malicious, our ensemble framework can still tolerate all the poisoning attacks across all robust

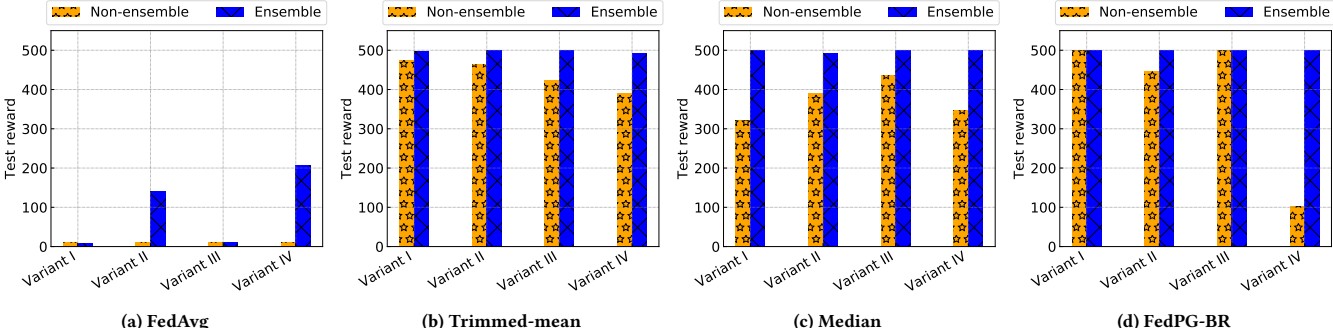

Figure 5: Different variants of our Normalized attack, where the Cart Pole dataset is considered.

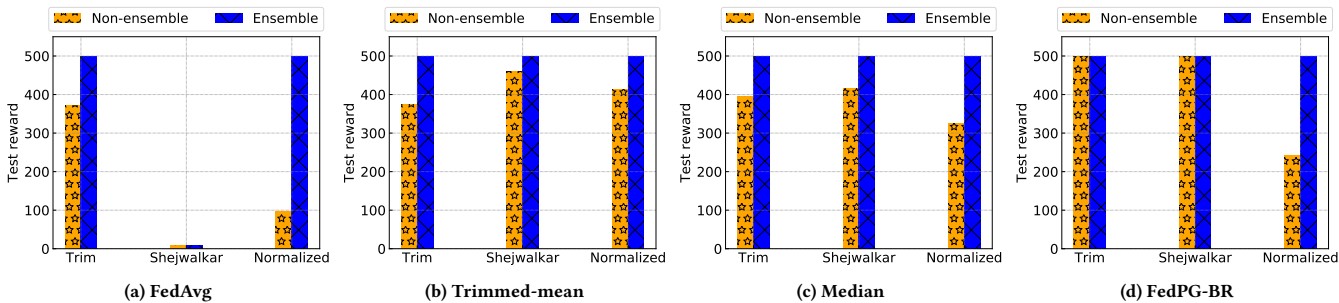

Figure 6: Results of partial knowledge attack, where the Cart Pole dataset is considered.

aggregation rules. For example, when 40% of agents are malicious, our ensemble method achieves similar test rewards with that of FedAvg without attack. However, as shown in Fig. 3, even when the fraction of malicious agents is 30%, existing robust aggregation rules under non-ensemble setting could be easily poisoned

**Impact of total number of agents:** Fig. 11 in Appendix shows the influence of varying total agent numbers on our ensemble method under various attacks, with the proportion of malicious agents set at 30% and the overall number of agents ranging from 30 to 90. The numbers of groups are set to 5, 7, 7, and 9 when the total agents are 30, 50, 70, and 90, respectively. We observe that our ensemble method remains robust when the total number of agents varies.

**Impact of number of groups:** Fig. 12 in Appendix shows the results of our ensemble method with different group numbers, with 30 agents, 30% of which are malicious. When there is only one group, the ensemble method becomes equivalent to the non-ensemble approach. For three Byzantine-robust methods under various attacks, the test rewards match those of FedAvg without attack when the group sizes are 3, 5, or 7.

Table 2: Different variants of our Normalized attack.

|  | Stage I | Stage II | Normalization |
|---|---|---|---|
| Variant I | ✓ | ✗ | ✓ |
| Variant II | ✗ | ✓ | ✓ |
| Variant III | ✓ | ✓ | ✗ |
| Variant IV (default) | ✓ | ✓ | ✓ |

**Impact of different perturbation vectors:** Our Normalized attack uses a perturbation vector $\Delta$. Table 6 in Appendix lists three types: "uv", "std", and "sgn". $\text{Avg}\{g_i : i \in [n]\}$ calculates the average of the $n$ local policy updates, and $\text{std}\{g_i : i \in [n]\}$ computes their standard deviation. "sgn" is our default perturbation vector. Fig. 13 in Appendix shows the results of FedAvg, Trimmed-mean, Median, and FedPG-BR under Normalized attacks with different vectors. "Normalized-uv" refers to the Normalized attack using the "uv" vector. We observe that FedPG-BR in the non-ensemble setting is particularly vulnerable to the "sgn" vector.

**Different variants of Normalized attack:** Our Normalized attack consists of two stages, with policy updates normalized during optimization. Table 2 outlines its variants. For example, Variant I skips Stage II, which optimizes the magnitude of malicious updates. Variant IV is our default attack. Fig. 5 shows the performance of these variants, where Variant IV demonstrates the most effective attack overall.

**Results of partial knowledge attack:** By default, we assume the attacker knows all agents' policy updates. Here, we explore a more realistic scenario where the attacker only knows updates from malicious agents. In this partial knowledge attack, we use $\text{AR}\{g_j : j \in \mathcal{B}\}$ to estimate the pre-attack aggregated update, where $g_j$ is the update from malicious agent $j$, and $\mathcal{B}$ is the set of malicious agents. Fig. 6 shows the results of the Trim, Shejwalkar, and our Normalized attacks. Even with partial knowledge, Byzantine-robust rules in non-ensemble settings remain vulnerable to poisoning. For example, FedPG-BR achieves a test reward of 242.6 under our Normalized attack.

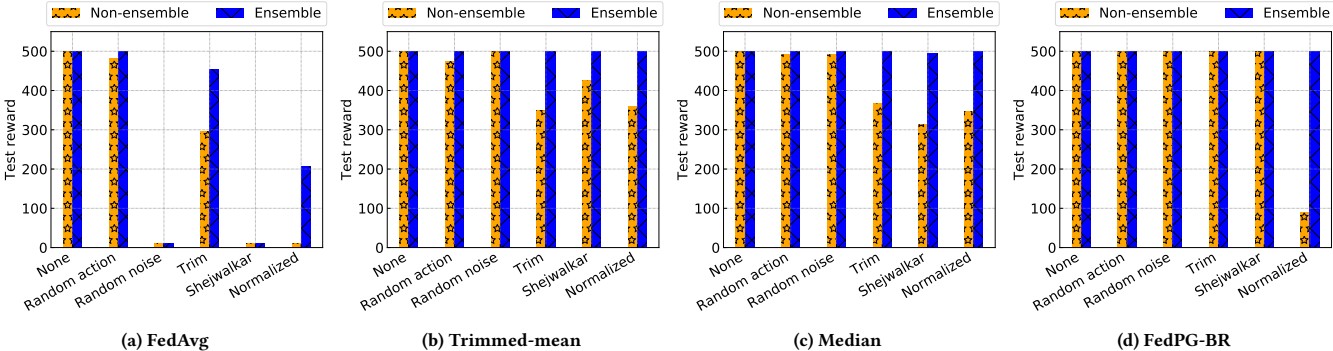

**(a) FedAvg**  **(b) Trimmed-mean**  **(c) Median**  **(d) FedPG-BR**

**Figure 7: Results of heterogeneous environment, where the Cart Pole dataset is considered.**

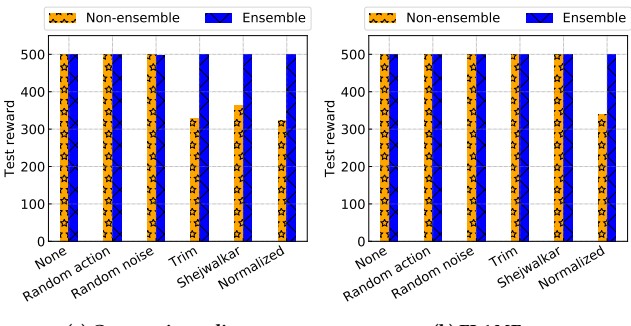

**(a) Geometric median**  **(b) FLAME**

**Figure 8: Results of geometric median and FLAME aggregation rules, where the Cart Pole dataset is considered.**

**Initiate the attack in the middle of the training phase:** We assume by default that the attacker targets the FRL system from the start of training. Here, we explore a scenario where the attack begins midway through training. In the Cart Pole dataset, each agent samples 5,000 trajectories during training. Fig. 14 in Appendix shows the results of FedPG-BR, Trimmed-mean, and Median in the non-ensemble setting. Results for the ensemble framework are omitted since it remains robust against all attacks, even if initiated from the start, as seen in Fig. 3. From Fig. 14, we observe that current robust aggregation methods in non-ensemble settings are still vulnerable to poisoning, even when the attack starts mid-training.

**Results of heterogeneous environment:** We explore a heterogeneous setting where two agents take the same action in the same state but receive different rewards. To simulate this, we add Gaussian noise $N(0, 0.1)$ to the rewards. Fig. 7 presents the results. We observe that non-ensemble aggregation rules remain vulnerable to poisoning attacks in heterogeneous environments. For example, FedPG-BR achieves a test reward of 100 under our Normalized attack. However, our ensemble framework effectively defends against all considered attacks using robust aggregation rules.

**Results of other foundational aggregation rules:** Fig. 8 shows results using geometric median [9] and FLAME [30] aggregation rules on the Cart Pole dataset. In our ensemble framework, the server still selects actions by majority vote since the action space is discrete. We observe that in non-ensemble settings, both geometric

median and FLAME are vulnerable to poisoning attacks. In contrast, our ensemble framework remains robust.

**Results of our ensemble method when using other aggregation rules to combine the $K$ continuous actions:** In our proposed ensemble framework, the server employs the geometric median aggregation rule to select the subsequent action during the testing phase when the action space is continuous. In this context, we investigate a scenario where the continuous actions are aggregated by the FedAvg or Trimmed-mean aggregation rules during the testing phase within our ensemble framework. The results are shown in Figs. 15-16 in Appendix, where the Inverted Pendulum dataset is considered (with a continuous action space). Figs. 15-16 show that robust aggregation rules like Trimmed-mean and Median remain vulnerable to poisoning when our ensemble framework uses FedAvg or Trimmed-mean for action prediction during testing. This occurs because FedAvg lacks robustness, and Trimmed-mean, as a coordinate-wise rule, can only filter individual outlier parameters, not entire malicious policy updates.

## 7 Conclusion, Limitations, and Future Work

We introduced the first model poisoning attacks on Byzantine-robust FRL. Rather than increasing the distance between the aggregated policy updates before and after the attacks, our introduced Normalized attack strives to amplify the angular deviation between the two updates. Additionally, we proposed a unique ensemble method that is provably resistant to poisoning attacks under some mild assumptions. Comprehensive experimental findings demonstrated that our Normalized attack can significantly corrupt Byzantine-robust aggregation methods in non-ensemble configuration, and our ensemble approach effectively safeguards against poisoning attacks.

A limitation of our work is that our Normalized attack requires the attacker to be aware of the server's aggregation rule. An intriguing avenue for future research would be to develop new attacks that do not necessitate such information. Our Normalized attack is limited to untargeted poisoning attacks, another interesting future work is to study targeted poisoning attacks [1, 39, 42] to FRL. Additionally, investigating security issues in multi-agent reinforcement learning [5, 21, 36, 38, 46] would be a fruitful area for further exploration.

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

# A Appendix

**Table 3: Architecture of MLPs for three datasets.**

| Parameter | Dataset | | |
|---|---|---|---|
| | Cart Pole | Lunar Lander | Inverted Pendulum |
| Hidden weights | 16, 16 | 64, 64 | 64, 64 |
| Activation | RELU | Tanh | Tanh |
| Output activation | Tanh | | |

## A.1 Proof of Theorem 1

Given a test state $s$, the action frequencies for actions $x$ and $y$ when up to $n'$ agents are malicious are represented as $v'(s, x)$ and $v'(s, y)$, respectively. Under the worst-case condition, for a specific group, if malicious agents are present, the global policy learnt by the group might predict action $y$ instead of $x$ at state $s$. That is, $v'(s, x)$ will decrease by 1 and $v'(s, y)$ will increase by 1 after the attack. Moreover, given that up to $n'$ agents can be malicious, a maximum of $n'$ groups may include malicious agents. Then we

---

**Algorithm 1** Training phase of our ensemble framework.

---

**Input:** Number of agents $n$; number of groups $K$; learning rate $\eta$; foundational aggregation rule $AR\{\cdot\}$; global training rounds $T$.
**Output:** Global policies $\theta_k^T, k \in [K]$.
 1: Divide $n$ agents into $K$ disjoint groups.
 2: Initialize $\theta_k^1, k \in [K]$.
 3: **for** $t = 1, 2, \cdots, T$ **do**
 4:     **for** each group $k \in [K]$ in parallel **do**
 5:         The server sends the global policy $\theta_k^t$ to all agents in group $k$.
 6:         **for** each agent $i \in n_k$ in parallel **do**
 7:             Updates $\theta_i^t$ and sends $g_i^t$ to the server.
 8:         **end for**
 9:         The server updates the global policy of group $k$ as $\theta_k^{t+1} \leftarrow \theta_k^t + \eta \cdot AR\{g_i^t : i \in n_k\}$.
 10:     **end for**
 11: **end for**

---

**Algorithm 2** Testing phase of our ensemble framework.

---

**Input:** State $s$, $K$ actions $F(s, \theta_k^T), k \in [K]$; action space $\mathcal{A}$.
**Output:** Action $\Phi(s)$.
 1: **if** $\mathcal{A}$ is discrete **then**
 2:     Computes action frequency for each action $a \in \mathcal{A}$ according to Eq. (10).
 3:     Obtains $\Phi(s)$ according to Eq. (11).
 4: **else if** $\mathcal{A}$ is continuous **then**
 5:     Calculates $\Phi(s)$ according to Eq. (12).
 6: **end if**

---

have that:

$$v'(s, x) \geq v(s, x) - n', \tag{16}$$

$$v'(s, y) \leq v(s, y) + n'. \tag{17}$$

In our proposed ensemble approach, when the test state $s$ is given, if the prediction of action $x$ still holds, then either Condition I or Condition II must be true:

$$\text{Condition I: } v'(s, x) > v'(s, y), \tag{18}$$

$$\text{Condition II: } v'(s, x) = v'(s, y) \text{ and } x < y, \tag{19}$$

where Condition II is true due to the assumption in Theorem 1 that if two actions possess the same action frequencies, the action with the smaller index is chosen.

Combining Eqs. (16)-(19), we have that:

$$n' \leq \left\lfloor \frac{v(s, x) - v(s, y) - \mathbb{1}_{\{y < x\}}}{2} \right\rfloor, \tag{20}$$

which completes the proof.

## A.2 Proof of Theorem 2

Let $F'(s, \theta_1^T), F'(s, \theta_2^T), ..., F'(s, \theta_K^T)$ be the set of $K$ actions after attack. Since $\Phi(s)$ and $\Phi'(s)$ are respectively the before-attack and after-attack aggregated policy updates, then $\Phi'(s) - \Phi(s)$ is the geometric median of $K$ vectors $\{F'(s, \theta_k^T) - \Phi(s) : k \in [K]\}$. Based on Lemma A.3, let $w$ be defined as $w = \max \left\{ \|F(s, \theta_k^T) - \Phi(s)\| : k \in [K] \right\}$,

and with the condition $0 < n' < K/2$, one has that:

$$\left\| \Phi(s) - \Phi'(s) \right\| \leq \frac{2w(K - n')}{K - 2n'}, \tag{21}$$

which completes the proof.

## A.3 Useful Technical Lemma

LEMMA 1. *Let's consider $v_1, \ldots, v_K$ to be $K$ vectors in a Hilbert space, let $v_*$ represent a $(1 + \epsilon)$-approximation of their geometric median. This means that for $\epsilon \geq 0$, we have $\sum_{k \in [K]} \|v_k - v_*\| \leq (1 + \epsilon) \min_z \sum_{k \in [K]} \|v_k - z\|$. Given any $r$ with the condition that $0 < r < K/2$ and a real number $w$, if the following condition satisfies:*

$$K - r \leq \sum_{k \in [K]} \mathbb{1}_{\|v_k\| \leq w}. \tag{22}$$

*Then one has:*

$$\|v_*\| \leq w\alpha + \epsilon\beta, \tag{23}$$

*where $\alpha = \frac{2(K-r)}{K-2r}$, $\beta = \frac{\min_z \sum_{k \in [K]} \|v_k - z\|}{K - 2r}$. Ideally, the geometric median sets $\epsilon = 0$.*

PROOF. This lemma is taken directly from [10, 26], so we omit its proof here. □

## A.4 Datasets

**Cart Pole [2]:** The Cart Pole environment is a simulation of a cart with a pole attached to it by a hinge. The cart can move along a horizontal track, and the pole can swing freely in the air. The goal is to balance the pole on the cart by applying forces to the left or right of the cart. The action space is a discrete space $\{0, 1\}$ representing the direction of the fixed force applied to the cart, where 0 for pushing the cart to the left, and 1 for pushing it to the right. A reward of +1 is added for every time step that the pole remains upright. The episode ends if the pole falls over more than 12 degrees from vertical, the episode length is greater than 500, or the cart moves more than 2.4 units from the center. Given that the maximum episode length is 500, the highest possible reward in this scenario should also be 500.

**Lunar Lander [11]:** The Lunar Lander environment is a simulation of a rocket landing on the moon. The rocket's engines are controlled by choosing one of four actions: do nothing, fire left engine, fire main engine, or fire right engine. The action space, therefore, is a discrete space and can be represented as $\{0, 1, 2, 3\}$. The goal is to land safely on the landing pad without crashing or going out of bounds. A reward is obtained for every step that the rocket is kept upright, and a penalty for using the engines. The environment is stochastic, meaning that the initial state of the rocket is random within a certain range.

**Inverted Pendulum [2]:** The Inverted Pendulum is similar to the Cart Pole problem, which is another classic control problem where you have to balance a pole on a cart by applying forces to the left or right. Yet, it is different from the Cart Pole in several key aspects. First, the Inverted Pendulum is powered by the Mujoco physics simulator[37], which allows for more realistic and complex experiments, such as varying the effects of gravity. Second, the action

space of the Inverted Pendulum is continuous. Thirdly, considering that the maximum episode duration is set at 1000, the utmost attainable reward for the Inverted Pendulum dataset is 1000.

## A.5 Compared Poisoning Attacks

**Random action attack [13]:** Random action attack is a category of data poisoning attacks in which malicious agents intend to corrupt their local trajectories. In particular, every malicious agent chooses a random action regardless of the state.

**Random noise attack [13]:** Random noise attack is a kind of model poisoning attack. In each training round, a malicious agent draws each coordinate of its policy update from an isotropic Gaussian distribution with a mean of 0 and a variance of 1,000.

**Trim attack [14]:** This attack operates under the assumption that the server uses Trimmed-mean [44] or Median [44] as its aggregation rule, to combine the local policy updates sent from agents. Trim attack considers each dimension of policy update independently. Specifically, malicious agents intentionally manipulate their policy updates so that the aggregated policy update post-attack differs significantly from the one before the attack, for each dimension of policy updates.

**Shejwalkar attack [34]:** In the Shejwalkar attack, the attacker designs malicious local policy updates with the intent to enlarge the distance between the aggregated policy update before the attack and the one after the attack.

## A.6 Foundational Aggregation Rules

**FedAvg [25]:** In FedAvg, once the server receives local policy updates from all agents, it calculates the global policy update by taking the average of these updates.

**Coordinate-wise trimmed mean (Trimmed-mean) [44]:** Upon receiving $n$ local policy updates, the server first discards the largest $c$ and smallest $c$ elements for each dimension, then computing the average of the remaining values, where $c$ is the trim parameter.

**Coordinate-wise median (Median) [44]:** In the Median aggregation rule, the server determines the aggregated global policy update by computing the coordinate-wise median from all received local policy updates.

**Geometric median [9]:** For the geometric median aggregation rule, the server computes the aggregated policy update by taking the geometric median of received local policy updates from all agents.

**FLAME [30]:** The FLAME method starts by computing the cosine similarity among agents' local policy updates. It then employs clustering methods like HDBSCAN [6] to identify potentially malicious updates. To further reduce the impacts of poisoning attacks, it implements an adaptive clipping mechanism to adjust the local updates. Finally, the server adds noise to the aggregated policy update to obtain the final global update.

**FedPG-BR [13]:** In the FedPG-BR aggregation rule, the server first calculates the vector median of all received local policy updates. A local policy update is deemed malicious if it fars from the calculated vector median. To additionally minimize the policy update variance, the server independently samples some trajectories to

compute a server policy update. Subsequently, the server leverages the Stochastically Controlled Stochastic Gradient (SCSG) [19] framework to update the global policy.

**Table 4: Additional parameter settings for three datasets.**

| Parameter | Dataset | | |
|---|---|---|---|
| | Cart Pole | Lunar Lander | Inverted Pendulum |
| $\gamma$ | 0.999 | 0.99 | 0.995 |
| $H$ | 500 | 1000 | 1000 |
| $\mathfrak{I}$ | 0 | | |
| $\Delta$ | $-\text{sign}(\text{Avg}\{\boldsymbol{g}_i : i \in [n]\})$ | | |
| $\hat{\lambda}$ | 0.83 (decays at each iteration with factor 1/3) | 1 (decays at each iteration with factor 1/3) | 0.83 (decays at each iteration with factor 1/3) |
| $\hat{\zeta}$ | 0.03 (decays at each iteration with factor 1/3) | 0.02 (decays at each iteration with factor 1/3) | 0.2 (decays at each iteration with factor 1/3) |

**Table 5: Parameter settings of FedPG-BR for three datasets.**

| Parameter | Dataset | | |
|---|---|---|---|
| | Cart Pole | Lunar Lander | Inverted Pendulum |
| $b$ | 4 | 8 | 12 |
| $N$ | $N \sim Geom(\frac{B}{B+b})$ | | |
| $\sigma$ | 0.06 | 0.07 | 0.25 |
| $\delta$ | 0.6 | 0.6 | 0.6 |

**Table 6: Different perturbation vector $\Delta$.**

| | uv | std | sgn (default) |
|---|---|---|---|
| $\Delta$ | $-\frac{\text{Avg}\{\boldsymbol{g}_i : i \in [n]\}}{\|\text{Avg}\{\boldsymbol{g}_i : i \in [n]\}\|}$ | $-\text{std}\{\boldsymbol{g}_i : i \in [n]\}$ | $-\text{sign}(\text{Avg}\{\boldsymbol{g}_i : i \in [n]\})$ |

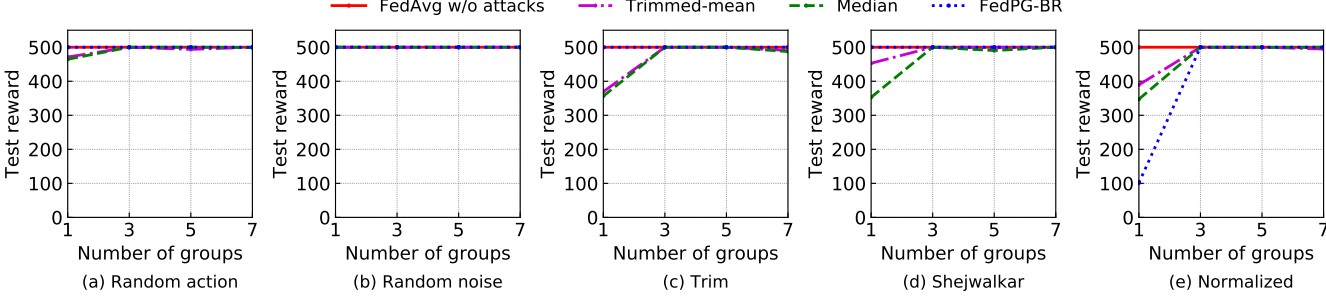

**Figure 9: Results on Lunar Lander dataset.**

**Figure 10: Results on Inverted Pendulum dataset.**

**Figure 11: Impact of the total number of agents on our ensemble method, where the Cart Pole dataset is considered.**

**Figure 12: Impact of the number of groups on our ensemble method, where the Cart Pole dataset is considered.**

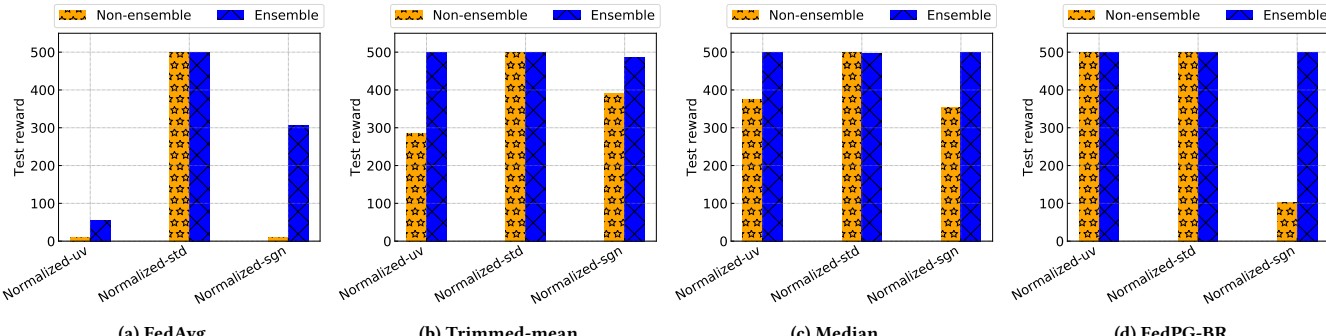

**(a) FedAvg**     **(b) Trimmed-mean**     **(c) Median**     **(d) FedPG-BR**

**Figure 13: Different perturbation vectors on our Normalized attack, where the Cart Pole dataset is considered.**

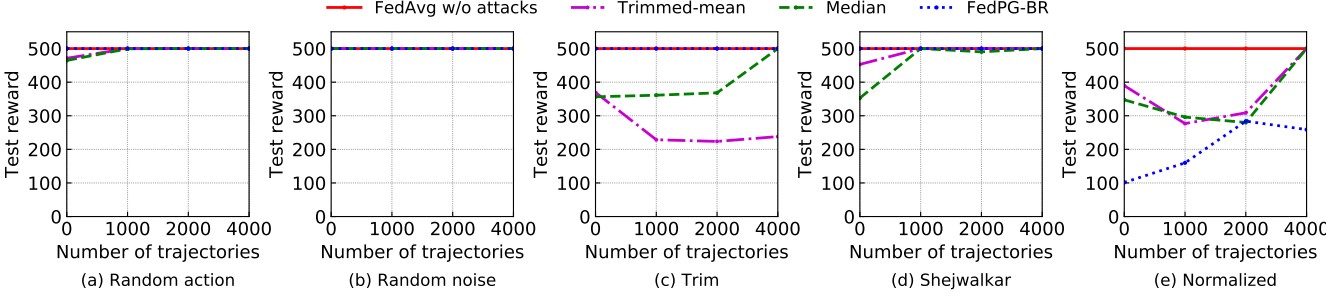

**(a) Random action**   **(b) Random noise**   **(c) Trim**   **(d) Shejwalkar**   **(e) Normalized**

**Figure 14: Impact of starting to attack after sampling a certain number of trajectories on different non-ensemble methods, where the Cart Pole dataset is considered.**

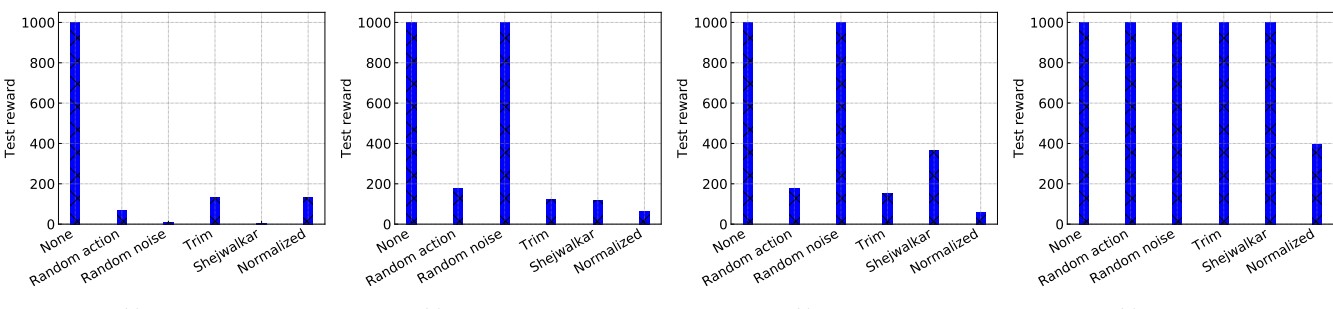

**(a) FedAvg**     **(b) Trimmed-mean**     **(c) Median**     **(d) FedPG-BR**

**Figure 15: Results of our ensemble method, where the continuous actions are aggregated by the FedAvg aggregation rule in the testing phase. The Inverted Pendulum dataset is considered.**

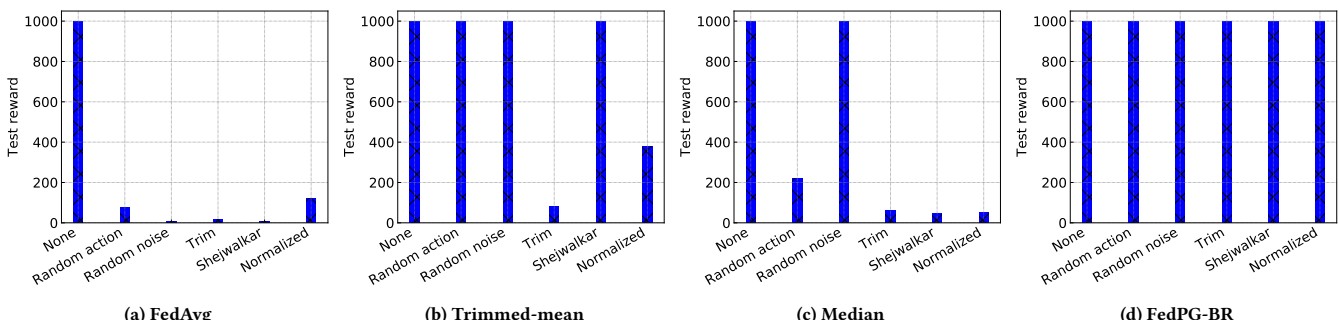

**(a) FedAvg**     **(b) Trimmed-mean**     **(c) Median**     **(d) FedPG-BR**

**Figure 16: Results of our ensemble method, where the continuous actions are aggregated by the Trimmed-mean aggregation rule in the testing phase. The Inverted Pendulum dataset is considered.**