# OpenReview forum: "Provably Robust Federated Reinforcement Learning"
_ACM.org/TheWebConf/2025/Conference — WWW 2025 Oral_

### Official Review · Reviewer_rh9F · 2024-11-04

**Novelty:** 3
**Technical Quality:** 3

**Review:**

This paper introduced a new Normalized attack for FRL and proposed an ensemble FRL approach that is provably secure against the introduced Normalized and current Byzantine attacks in FRL. There are already lots of robust algorithms proposed in federated learning, and the proposed solution is not novel enough. Some detailed comments are provided as follows.

Strengths:
1. The paper introduced a new attack and proposed a corresponding solution.
2. The experiments look sufficient.


Weaknesses:

1. The topic of federated learning is very mature, and there are already lots of papers that have studied robust aggregation methods against Byzantine attacks. Therefore, the contribution of the technique is not novel enough.

2. The depth of the paper is not good enough because it focuses on two purposes: the attacking and the defense. Using 8 pages to write two topics is usually not deep as using 8 pages to focus only on one problem.

3. The majority vote ensembling for robust federated learning is not novel; there are already papers in 2019 and 2021 that have proposed similar ideas [1][2], which have not been referred to in this paper.

4. In Figure 4, we can see that the proposed ensemble scheme works when the malicious agents' fraction is $50\%$. We know the ensemble scheme is based on a majority vote; how does it work when 50% of agents are malicious?




[1]. Ozdayi, Mustafa Safa, Murat Kantarcioglu, and Yulia R. Gel. "Defending against backdoors in federated learning with robust learning rate." Proceedings of the AAAI Conference on Artificial Intelligence. Vol. 35. No. 10. 2021.

[2]. Bernstein, Jeremy, et al. "signSGD with majority vote is communication efficient and fault tolerant." ICLR 2019

**Questions:**

See Weaknesses.

**Reviewer Confidence:**

4: The reviewer is certain that the evaluation is correct and very familiar with the relevant literature

**Scope:**

3: The work is somewhat relevant to the Web and to the track, and is of narrow interest to a sub-community

---

### Official Review · Reviewer_9eoe · 2024-11-11

**Novelty:** 6
**Technical Quality:** 6

**Review:**

**Paper Summary**
This paper introduces the first model poisoning attacks against Federated Reinforcement Learning (FRL) and additionally develops a provably secure ensemble FRL approach.

**Review Summary**
I found the ideas presented in this paper to be novel and interesting.
Overall, I believe the paper makes good contributions, but I hope to have some of my questions answered as well.

**Detailed Comments**
I have one major comment, which I believe will substantially improve the paper.

1. Motivation. I believe what is currently missing from the paper is a more in-depth discussion on the motivation for model poisoning attacks. For instance it is currently unclear _why_ agents might want to carry out such an (untargeted) poisoning attack with respect to real-world deployments of FRL. Furthermore, it also seems unlikely that malicious agents could have access to even other agents' policy updates when we are working precisely under the FL setting where other agents' data is meant to be private. Therefore, I would recommend the authors to motivate the problem and threat model more substantially.

**Questions:**

1. FedPG-BR also proves byzantine robustness against 50% of malicious agents. However, your normalized attack successfully reduced the test reward significantly with just 30% of malicious agents. This seems like a significant result, but I can't find much explanation why. Is their proof wrong or does it not cover your threat model?

2. Your normalized attack is meant to optimize the angular deviation between the benign and malicious updates. However it seems like FLAME also looks at the cosine similarity, which depends on the angle. So why does your attack work against FLAME? (I am assuming its the normalization?)

**Reviewer Confidence:**

3: The reviewer is confident but not certain that the evaluation is correct

**Scope:**

4: The work is relevant to the Web and to the track, and is of broad interest to the community

---

### Official Review · Reviewer_FNid · 2024-11-16

**Novelty:** 3
**Technical Quality:** 3

**Review:**

# Quality:
The paper is well-structured and comprehensive experiments. However, it lacks critical theoretical analysis, and the experimental design and presentation suffer from issues in adherence to standards, affecting its credibility.

# Clarity:
The paper provides detailed descriptions of the theory and experiments, but insufficient explanations for key phenomena make some aspects unconvincing.

# Originality:
The proposed normalized attack is designed to exploit the unique characteristics of FRL, demonstrating a certain level of innovation, but the core idea remains relatively simple with limited breakthrough impact.

# Significance:
The work addresses the significant problem of FRL security, which is highly relevant. However, the limitations in experimental results reduce its applicability to real-world scenarios.


# Pros
1. The paper clearly articulates the differences between attacking FRL and FL, providing a reasonable explanation for the need to design dedicated attacks for FRL.
2. The proposed attack method effectively targets FRL, covering both common robust aggregation algorithms and the defense methods proposed by the authors.

# Cons
1.  The authors place critical experimental results or supporting arguments in the appendix, while extensively discussing these results in the main text. This approach is unfair to other submissions and does not comply with the standards of WWW.

2.  The paper employs basic robust aggregation methods without comparing them against current SOTA methods (e.g., more advanced robust aggregation rules like DnC by Shejwalkar or FLTrust). This inadequacy undermines the evaluation of the effectiveness of the proposed defense method. Furthermore, the attacks mentioned in the paper can bypass the defense proposed by the authors.

3.  The core idea of the normalized attack is to measure angular distance rather than Euclidean distance. However, as far as I know, Shejwalkar's Min-Max attack already possesses this capability and can effectively poison angular-distance-based defenses such as cosine similarity. In comparison, the normalized attack appears relatively straightforward, lacking a paradigm-shifting innovation.

4.  The normalized attack lacks sufficient theoretical analysis, resulting in inadequate alignment between the theory and experiments to convince the reader.

5.  In Theorem 2, the authors mention that the proportion of malicious agents should be less than 50%. However, in Figure 4, when the proportion reaches 50%, the rewards for other comparison methods drop to 0, while the proposed method maintains results identical to those without any attack. This inconsistency is puzzling and lacks a reasonable explanation.

**Questions:**

1. Figures 3 and 4 indicate that FedPG-BR can completely defend against current SOTA poisoning attacks (including the normalized attack). This result is surprising. As far as I understand, Shejwalkar's Min-Max attack can achieve the same distance while reversing the direction, which seems similar to the core principle of the normalized attack. Could you explain in more detail how your method achieves complete defense in this scenario?

2. Additionally, in some experiments, you assume the proportion of malicious clients is as high as 50%. Under this condition, the training data available to the system is significantly reduced. Yet, your results show test accuracy identical to FedAvg without attacks. This is particularly concerning. Could you elaborate on the experimental setup and provide a detailed analysis to help understand the rationale behind these results?

**Reviewer Confidence:**

4: The reviewer is certain that the evaluation is correct and very familiar with the relevant literature

**Scope:**

4: The work is relevant to the Web and to the track, and is of broad interest to the community

---

### Official Review · Reviewer_syHo · 2024-12-01

**Novelty:** 4
**Technical Quality:** 4

**Review:**

This paper proposes a robust model attack on federated reinforcement learning (FRL), called "normalization attack". The attack destroys the robustness defense mechanism of FRL system by maximizing the angular deviation between the aggregated policy updates before and after the attack, rather than just increasing the distance. To deal with this attack, the paper proposes an ensemble learning-based method

**Strengths:**

+This paper is the first model poisoning attacks tailored to Byzantine-robust FRL.

+A large number of experiments and sufficient theoretical proofs have been used to illustrate the effectiveness of the proposed method.

+This paper is well-organized and easy to follow.

**Weaknesses:**

-The article assumes that the attacker has the server rules. Whether this assumption is reasonable needs to be considered. Generally speaking, the attacker can control some malicious clients but does not know any knowledge of the server.

-The communication and computational overhead of the defense method proposed in the article is not discussed.

-The article only considers non-target attacks. Adding targeted attacks will make the article more valuable.

**Questions:**

See the weaknesses.

**Ethics Review Flag:**

Yes

**Reviewer Confidence:**

3: The reviewer is confident but not certain that the evaluation is correct

**Scope:**

3: The work is somewhat relevant to the Web and to the track, and is of narrow interest to a sub-community

---

### Official Review · Reviewer_ytEg · 2024-12-02

**Novelty:** 6
**Technical Quality:** 5

**Review:**

Summary:
This paper introduced a new attack method for Federated reinforcement learning, **Normalized attack**. To counter this and other attacks, the authors propose an ensemble FRL approach that trains multiple global policies using different agent groups, and the final action is determined by majority vote or geometric median. The authors also proposed firm experimental results, indicating that normalized attack can significantly disrupt existing Byzantine-robust methods.

Pros:
- The paper introduces a new attack strategy, the "Normalized attack," which focuses on maximizing the angular deviation between pre- and post-attack aggregated policy updates. This is a distinct approach compared to prior methods.
- The proposed ensemble FRL method is backed by formal proofs of robustness against poisoning attacks under specific conditions.
- The authors treat the attack as an optimization problem based on normalized distance before and after policy aggregation. Besides, they solve the problem in a two-stage manner.
- The authors also propose defense method against the proposed attack method through emsembling training multiple global policies.
- The authors proposed theoretical security analysis for both descrete and continous action space.
- Extensive trials on several benchmark datasets (e.g., Cart Pole, Lunar Lander, Inverted Pendulum) illustrate the efficacy of both the Normalized assault and the ensemble defensive method. The results encompass comparisons with cutting-edge aggregation rules.



Cons:
- The suggested ensemble method elevates computational and communication burdens owing to the necessity of training many global policies and including extra aggregation procedures during inference.
- The attack is primarily focused on untargeted poisoning. A broader exploration of targeted attacks could further strengthen the study.

**Questions:**

Here are the questions:
- When the number of groups is big, how does the ensemble approach fare perhaps resulting in higher computing and communication overhead?
- Can the suggested Normalized attack be expanded to situations were the attacker only partially knows the aggregation rule?
- The ensemble approach addresses dynamic situations in which the number of hostile actors or their conduct varies with time.
- To assess practical viability and robustness, have you thought about testing the framework in actual applications (such as robots or IoT networks)?

**Reviewer Confidence:**

3: The reviewer is confident but not certain that the evaluation is correct

**Scope:**

3: The work is somewhat relevant to the Web and to the track, and is of narrow interest to a sub-community